# Antitumor Effect of Bleomycin Nanoaerosol in Murine Carcinoma Model

**DOI:** 10.3390/molecules28104157

**Published:** 2023-05-18

**Authors:** Saida S. Karshieva, Gulalek Babayeva, Vadim S. Pokrovsky, Yuri M. Shlyapnikov, Elena A. Shlyapnikova, Anna E. Bugrova, Alexey S. Kononikhin, Evgeny N. Nikolaev, Igor L. Kanev

**Affiliations:** 1Laboratory of Biochemical Pharmacology and Cancer Models, N.N. Blokhin National Medical Research Center of Oncology, 115478 Moscow, Russia; skarshieva@gmail.com (S.S.K.); babaeva-g@rudn.ru (G.B.); v.pokrovsky@ronc.ru (V.S.P.); 2Research Institute of Molecular and Cellular Medicine, Patrice Lumumba People’s Friendship University, 117198 Moscow, Russia; 3Center of Genetics and Life Sciences, Sirius University of Science and Technology, 354340 Sochi, Russia; 4Institute of Theoretical and Experimental Biophysics, Russian Academy of Science, 142290 Pushchino, Moscow region, Russia; 5Emanuel Institute for Biochemical Physics, Russian Academy of Science, 119334 Moscow, Russia; 6Center of Life Science, Skolkovo Institute of Science and Technology, 121205 Moscow, Russia; a.kononikhin@skoltech.ru

**Keywords:** lung cancer, nanoaerosol, nanoparticles, bleomycin, pharmacokinetics, drug delivery

## Abstract

Bleomycin, which is widely used as an antitumor agent, possesses serious adverse effects such as pulmonary toxicity. Local nanoaerosol deposition for lung cancer treatment is a promising alternative to drug delivery to lung lesions. The aim of this work is to test the hypothesis that bleomycin nanoaerosol can be effectively used to treat multiple lung metastases. To obtain bleomycin nanoaerosol, an aerosol generator based on electrospray of a solution of a nonvolatile substance with gas-phase neutralization of charged aerosol particles was used. Lung metastases in murine Lewis lung carcinoma and B16 melanoma animal models were counted. The effect of inhaled bleomycin nanoparticles on the number and volume of metastases, as well as pulmonary side effects, was investigated. Using a mouse exposure chamber, the dose-dependent effect of inhaled bleomycin on tumor volume was evaluated in comparison with intraperitoneal administration. Bleomycin nanoaerosol reduced the volume of metastases and produced a higher antitumor effect at much lower doses. It has been established that long-term exposure to nanoaerosol with a low dose of bleomycin is capable of suppressing cancer cell growth. The treatment was well tolerated. In the lungs, minor changes were found in the form of focal-diffuse infiltration of the lung parenchyma.

## 1. Introduction

Advanced lung cancer has one of the highest mortality rates among all common cancers, and patients have an average life expectancy of less than 5 years. Lung cancer and lung metastasis require the use of combinations of potent chemotherapeutic agents. Over the past decade, a number of reviews have been published on improving the effectiveness of treatment, including a targeted delivery to the lungs for drugs, genes, proteins, and enzymes [1,2,3,4,5,6]. The emergence of the COVID-19 pandemic has sparked a significant increase in interest in the aerosol route for antiviral drug delivery [7,8] and vaccination [9]. Local delivery via inhalation is a suitable alternative to provide higher local drug concentration. Another advantage of the inhalation route of administration is that it enables the delivery of low doses of inhaled drugs to the lungs while significantly reducing toxic effects [10].

It should be emphasized that a common approach of drug delivery is inhalation using a nebulizer, which gives an aerosol with a wide range of particles. However, even relatively mild ultrasound nebulizers are capable of damaging fragile biomolecules and drugs [11]. For lung cancer treatment, a certain range of nanoparticle size is crucial because particles of different sizes settle differently in parts of the respiratory tract [12,13,14]. The probability of deposition in the alveoli decreases as particle size increases and is relatively low for particles larger than 5 µm. Nanoparticles (NAPs) smaller than 10 nm are likely to be eliminated by the reticuloendothelial system; 30% of NAPs of 20 nm diameter accumulate in the nasopharyngeal and tracheobronchial area, and approximately 50% of them in the alveolar area [14]. Moreover, after sedimentation in the lungs, NAPs with a diameter of more than 100 nm [5] are consumed by alveolar macrophages. It has been shown that NAPs between 10–100 nm penetrate deeply into the lungs and settle in the alveoli [15]. Various methods have been proposed for the preparation of NAPs for inhalation for efficient delivery of pure drugs as well as drug-containing polymeric micelles [16,17], polymeric NAPs [18,19], and lipidic and inorganic nanocarriers [20,21,22,23,24]. Recently, the technology of nanoaerosols has been developed using an evaporation–condensation aerosol generator, and drug delivery by this route demonstrated high bioavailability and significantly reduced the required dose of inhaled drugs compared with delivery by the oral route [15,25,26]. Using a custom nanoaerosol generator designed to atomize biological and biologically active substances, we showed that iron oxide NAPs obtained by the electrospraying of a suspension with gas-phase neutralization with an average particle size of about 100 nm penetrates deeply into the lungs [27]. The electrospray-based atomization process is safe for the structure and functional activity of biological substances [28]. We also demonstrated that nanoaerosols reduce the required effective dose of liposomal levofloxacin against pulmonary murine Francisella tularensis subsp. novicida infection [29]. The analysis of quantum dot distribution in cryosections of murine lungs showed that nanoaerosol particles penetrate the alveoli and spread more homogeneously in lungs than upon intranasal delivery (ibid.).

Bleomycins (BLM) are a family of non-ribosomal glycopeptides which were first isolated from the actinomycete *Streptomyces verticillus* [30]. BLM is clinically used as an antitumor antibiotic in the treatment of various cancers, including Hodgkin’s and non-Hodgkin’s lymphomas, ovarian cancer, melanoma, sarcoma, and squamous cell cancers [31]. BLM has also long been used in lung cancer chemotherapy [32]. However, it was reported that intravenous administration of BLM is associated with pulmonary fibrosis, and this effect can be reproduced in experimental animals [33,34,35,36]. Similar findings have been described in mice that received BLM both intravenously and intratracheally [37,38]. The occurrence of pulmonary fibrosis is explained by the stimulatory effect of BLM on the synthesis of TIMP-1, INF-γ, MMP9, and TGF-β [39]. Suppression of this effect protects against pulmonary fibrosis caused by BLM [40,41]. A recent study demonstrated that the inhibition of Toll-like receptor signaling also has protective effects in the BLM-induced lung fibrosis model [42]. As we previously showed [38], slow administration of ~100 nm NAPs of BLM for 13 days with a total dose of ~1 mg/kg did not cause pneumofibrosis in lungs, in contrast to bolus instillation of similar doses; only pneumonitis and infiltration of macrophages in all parts of the lungs were observed. These effects were notably reversible and disappeared after two weeks without aerosol inhalation. The aim of this work was to try the treatment of lung cancer with BLM nanoaerosol in the murine carcinoma model.

In the current study, we applied the electrospray-based nanoaerosol generator [28,29,43] to deliver ~100 nm BLM NAPs into mouse lungs, and compared the action of nanoaerosol particles of BLM with intraperitoneal administration. The aim was to determine whether such sub-toxic doses are still capable of arresting cancer cell growth when introduced slowly. As a result, it was shown that BLM in nanoaerosol form reduces the volume of metastases and acts in much lower doses.

## 2. Results

### 2.1. Dose Calculation

BLM aerosol concentration at the entrance to the mouse chamber was evaluated using three different methods in a preliminary experiment. The total concentration of aerosol particles obtained using a scanning mobility particle sizer was ~10^8^ particles per L. The mass concentration of particles with a size of up to 300 nm, calculated from the spectrum, was 0.32 μg/L. The average concentration of BLM nanoaerosol obtained by using the nylon nanofilter was 0.23 μg/L. The difference can be explained by the variability of aerosol concentration over time. To calculate the final inhaled dose, we used the amount of sprayed BLM obtained using the nanofilter, since it characterizes the integral dose over the entire exposure time. The mass concentration of glucose aerosol particles with a size of up to 300 nm, calculated from the spectrum, was 0.34 μg/L.

The correction of the inhaled aerosol concentration for the deposition of aerosol in the mouse chamber was estimated in a preliminary experiment. The total mass concentration of aerosol measured using a piezobalance dust monitor at the exit from the chamber was 60% of the concentration measured at the entrance, which can be explained by the deposition of aerosol inside the chamber. Considering that mice could move inside the chamber, an average concentration of aerosol consumed by mice can be estimated as an arithmetic average of aerosol concentrations at the entrance and at the exit. Therefore, an adjustment for aerosol deposition was made in further calculations of the inhaled dose, which decreased the initial concentration by a factor of (1 + 0.6)/2 = 0.8.

The results of the aerosol concentration measurements and expected inhaled doses of BLM for all experiments are presented in Table 1. The results of the corresponding daily measurements of aerosol concentration and expected inhaled doses of BLM in all experiments are presented in the Appendix A.

This section may be divided by subheadings. It should provide a concise and precise description of the experimental results and their interpretation, as well as the experimental conclusions that can be drawn.

### 2.2. Antitumor Effect of NAPs on the Murine Melanoma Model

We evaluated the effect of NAPs of BLM on the growth of lung metastasis of metastatic murine B16F10 melanoma. On day 4 after s.c. implantation of melanoma, mice were randomly divided into two groups (n = 10 for each group) as follows: Group 1, exposed to BLM NAPs (experiment 1 in Table 1); and Control group 2, exposed to glucose NAPs. We observed (Figure 1A,B) that the dynamics of tumor growth in the control group were characterized by an exponential curve, and in the group with BLM, tumor growth was inhibited. NAPs inhibited tumor growth from 10 to 20 days after the start of treatment, and TGI reached maximum of 82% (*p* < 0.01) and increased the life span of mice by 26% compared with the control group (*p* = 0.035). These results indicate that NAPs had a significant inhibitory effect on melanoma tumor growth in vivo. No severe toxic effects (weight loss or sudden death) were associated with BLM NAP treatment. However, the difference in the number of spontaneous lung metastatic lesions between the experimental and control groups was insignificant (Figure 1C).

### 2.3. Comparative Effectiveness of NAPs and Intraperitoneal Injections of BLM

To evaluate the benefits of BLM NAPs, we compared their antitumor activity with systemically administered BLM in mice bearing melanoma B16F10. C57Bl6 mice grafted with B16F10 cells were divided into five groups (n = 8 for each group) as follows: Group 1, exposed to BLM NAPs for 5 h daily for 14 days (experiment 2 in Table 1); Group 2, exposed to BLM NAPs for 2.5 h daily for 14 days (experiment 3 in Table 1); Group 3, injected with 8 mg/kg BLM i.p.; Group 4, injected with 4 mg/kg i.p.; and Control group 5, injected with 0.9% NaCl i.p. A small tumor mass of 5.11 ± 1.27 mm^3^ was palpable at day 5 post-implantation, when we started treatment. Tumors in the control groups grew rapidly, reaching an average volume of 490.7 ± 88.8 mm^3^ by day 15 and 2759.2 ± 444.1 mm^3^ by day 22 after inoculation of B16F10 melanoma tissue suspension. In contrast, TGI in Group 3 was 72% (*p* = 0.002) and 66% (*p* = 0.003) by the same days, with tumor volume remaining at an average of 137.9 ± 43.6 mm^3^ and 952.8 ± 197.7 mm^3^, respectively (Figure 2A). The efficacy of BLM at a dose of 4 mg/kg was lower: TGI was 52.3% (*p* = 0.028) and 58.7% (*p* = 0.015) on days 10 and 17, respectively, after starting of treatment. In groups with 5 h exposure to BLM NAPs, the maximum effect of 53.1% (*p* = 0.05) was reached on day 14. No significant differences between the treated groups were observed. On day 23 after melanoma transplantation, all mice were euthanized and autopsied, and the lungs, spleens, and livers were harvested and weighted. There were no differences in the numbers of spontaneous pulmonary metastases between the control and treated groups, but we found a twofold decrease in average spleen weight in the groups treated with 4 mg/kg (0.104 ± 0.008 g vs. 0.226 ± 0.048 g, *p* = 0.017) and 8 mg/kg BLM (0.107 ± 0.011 g vs. 0.226 ± 0.048 g, *p* = 0.022) compared to the control group (Table 2). A significant decrease in relative spleen weight was also observed after treatment of mice with BLM at doses of 4 mg/kg and 8 mg/kg compared with the control group (0.42% and 0.44% versus 0.75%, respectively) (Appendix A). Other mice organs showed no visible pathological findings.

### 2.4. Antitumor Effect of NAPs on the Murine Lewis Lung Carcinoma Model

Another model studied was the metastatic murine model of lung cancer. Mice were randomly divided into three groups (n = 10): one group was exposed to BLM NAPs for 5 h daily over 14 days, one group was treated with i.p. 4 mg/kg BLM, and a control group was treated with i.p. 0.9% NaCl in the same treatment schedule. The mice were sacrificed on day 23 after tumor transplantation; lungs were weighed and fixed for histological analysis. In the control group, the tumor size increased nearly exponentially from day 6 and continued to increase rapidly, reaching a volume 4033.96 ± 475.00 mm^3^ by day 22 (Figure 3). BLM i.p. treatment showed a TGI of 52.7% (*p* = 0.001) and 71.2% (*p* < 0.01) compared with the control on days 10 and 17 after the beginning of treatment, respectively. BLM NAPs were less effective than BLM i.p. and inhibited tumor growth by 55.4% (*p* = 0.001) and 41.3% (*p* = 0.023) compared with the control on the same days, however their effect is clear.

### 2.5. Histological Analysis of the Lungs

Groups of five mice were used. Group 1 was exposed to BLM NAPs with the lung-deposited dose of 8.5 μg/mouse or 0.35 mg/kg. Group 2 inhaled 1% glucose NAPs during the same time, and group 3 was untreated. Mice from group 4 received a single intratracheal injection of 50 µL of BLM (5 mg/kg). The mice were euthanized immediately after the end of treatment. Five stained lung sections were viewed at a minimum. The lungs of animals from groups 1–3 had a normal structure without significant destructive pathologic changes (Appendix A). In group 1, only focal-diffuse infiltration of the lung parenchyma by interstitial macrophages and mononuclear cells was observed. Pneumofibrosis was detected in two animals from group 4.

### 2.6. Pharmacokinetics of BLM NAPs

For measuring low serum BLM concentrations, an advanced LC-MS/MS-based protocol was developed with a detection limit of 10 ng/mL. Groups of mice (n = 5) were sacrificed after 3, 10, 30, 60, and 120 min of BM NAP treatment and 10, 30 and 60 min after treatment was stopped. According to the MPPD model, each mouse received 300 ng of BLM in two hours. To compare the administration routes, mice were sacrificed 2, 5, 10, 30 and 60 min after injection of 4 mg/kg BLM. Serum analysis showed that, despite large variations in BLM concentration, it tended to increase with increasing inhalation time and reached ~100 ng/mL after 2 h (Figure 4). This is consistent with a delivered BLM dose of 300 ng, corresponding to a concentration of 150 ng/mL given 2 mL of mouse blood. The rapid clearance of BLM NAPs is also confirmed by a rapid (2–3 times in 10 min) drop in its concentration after inhalation was stopped (Figure 4A). In contrast, the half-life of the injected BLM was about 0.5 h (Figure 4B). This demonstrates a high bioavailability of BLM nanoaerosol, which quickly enters the bloodstream from the lungs.

## 3. Discussion

Local nanoaerosol deposition for lung cancer treatment is a promising approach for delivering drugs to lesions in lungs. An advantage of pulmonary delivery is that localized effect requires lower doses of inhaled drugs and therefore any toxic effect on organs or healthy cells is significantly reduced.

The first chemotherapeutic drug tested as an inhaled nanomedicine for the treatment of primary lung cancer, as well as pulmonary metastases, was liposomes containing a 9-nitro-20(S)-camptothecin topoisomerase inhibitor [44]. It was shown that the inhaled doses necessary to obtain similar plasma levels were four times lower than those administered orally. As a result, partial remission was observed. To study the effectiveness of inhaled antitumor NAPs for lung cancer treatment, mouse models are widely used. Significant improvement of antitumor efficacy and survival rate was demonstrated by the action of doxorubicin-loaded NAPs in the lung tumor orthotopic murine model (H460 cell line) [45]. A sixfold decrease in tumor volume was observed in the case of application of gelatin NAPs loaded with cisplatin using liquid nebulizer in lung tumor orthotopic and subcutaneous murine models (A549 cell line) [46]. The use of paclitaxel-loaded solid lipid NAPs for lung tumor therapy by inhalation in murine M109-HiFR lung carcinoma significantly increased antiproliferative activity compared with Taxol [47]. In our investigation, pure BLM NAPs without any additives were used, and, as previously shown, the nanoaerosol form of this drug does not cause permanent lung damage. BLM NAPs were produced using an aerosol generator, which enabled the long-term stable generation of nanoaerosols without damaging the sprayed substance molecules [27,28,29]. Its operation is based on electrohydrodynamic spraying of water and ethanol solutions followed by gas-phase neutralization. Nanoaerosol particles (NAPs) generated by this technique bear a few charges on each particle [48]. Earlier, we showed that charged NAPs were deposited within a dry swine lung model twice as efficiently as neutral NAPs of the same size [48]. This is in good agreement with the data of Lee et al. [5]: NAPs with positive charges had a higher affinity for tumor cells and could more easily penetrate them. In addition, a particle size of 10–100 nm was found to effectively target a tumor in an animal model. Here, we applied this generator to deliver BLM NAPs of ~100 nm in diameter into mouse lungs, and compared the action of nanoaerosol particles of BLM with intraperitoneal administration. We tried to determine whether such sub-toxic doses are still capable of arresting cancer cell growth when introduced slowly. In total, four experiments on the treatment of mice with BLM nanoaerosol for 14 days were conducted (see Table 1). Therefore, for the entire experiment, each mouse received only about 8 µg of BLM, or 0.35 mg/kg. This is much less than the amount used for intraperitoneal injections (4 and 8 mg/kg). This likely explains the increase in the size of the spleen in the latter case. As follows from the results, the scatter of the dose values is significant. This may be due to the variability of aerosol concentration in time caused by electrostatic phenomena. It was shown that BLM in nanoaerosol form reduces the volume of metastases and acts in much lower doses, producing promising prospects of improvement in the tolerability of BLM treatment. At the same time, the dose of 0.35 mg/kg is comparable to that in our previous study [38], which demonstrated no significant toxic effects of BLM NAPs on healthy mice. Together with data in Table 2, demonstrating no hematological toxicity of the BLM nanoaerosol dose, this comparison gives evidence that cancer treatment by BLM nanoaerosol without profound side effects was achieved in the present work. A histologic analysis was performed to observe possible lung abnormalities and pulmonary fibrosis after the aerosol treatment. From the toxic effect of NAPs, only focal-diffuse infiltration of the lung parenchyma by interstitial macrophages and mononuclear cells without obvious signs of pulmonary fibrosis were revealed.

The main advantage of the nanoaerosol drug form for lung treatment is its targeted delivery. Here, a comparable treatment effect of BLM was reached with ~10 times lower dose of drug, as compared with intraperitoneal administration. In the latter case, the administered drug enters the circulatory system, passes the liver, and only then reaches the lungs. At the same time, the drug delivered directly into the lungs in nanoaerosol form avoids its distribution across the whole organism and metabolism in the liver.

Yet another important difference between groups with a single intraperitoneal administration and groups treated with BLM aerosol is the different kinetics of the concentration in the blood and lung tissues. According to [49], after intratracheal administration, BLM can be found in the lungs for a rather long time (elimination half-time is 32 min) and demonstrates first-order kinetics of elimination. According to the same paper, after a single subcutaneous injection, the maximum concentration level was observed between 45–60 min after injection and was less than 1% of the administered dose. Thus, it is expected that, with inhaled BLM, the concentration of BLM in the lung tissues reached an equilibrium level and remained constant for the rest of the time of the experiment every day, and with intraperitoneal administration of BLM, a single high concentration peak is expected, followed by a rather rapid decrease (the half-time of BLM removal for C57Bl/6N mice after intravenous administration is 9.6 min [50]).

## 4. Materials and Methods

### 4.1. Reagents

BLM sulfate from *S. verticillus* (Nippon Kayaku Co., Ltd., Tokyo, Japan) was acquired from a local pharmacy. Glucose, nylon, and ethanol were purchased from Sigma-Aldrich (St. Louis, MO, USA).

### 4.2. Electrospray-Based Nanoaerosol Generator

The experimental setup is presented in Figure 5. BLM nanoaerosol was generated as described previously [27,38]. The principle of operation of the generator is the electrospraying of a solution with the subsequent neutralization of oppositely charged aerosol particles in the gas phase. Briefly, a BLM solution is placed in a capillary to which a high voltage is applied, resulting in the ejection of a cloud of highly charged microdroplets. The droplet size quickly decreases in a series of electrostatic decays caused by repulsion of charges in the drying microdroplets. The cloud of highly charged dry residues are neutralized with counterions and oppositely charged nanoparticles produced by electrospraying with another polarity, yielding a neutral or low charged nanoaerosol. A metal generator chamber with dielectric inserts forming an ellipsoidal inner space with round grounded electrode between inserts, was used. The generator design is described in more detail in Appendix A and in the patent [51]. A 0.5% solution of BLM sulfate in 50% (vol.) ethanol was sprayed simultaneously from capillaries with a positive and a negative potential. The values of the currents applied to the positive and negative capillaries were 90 and 40 nA, respectively, and the rate of the air flow through the generator chamber was 2.1 L/min. In control experiments under the same conditions, a 1% glucose solution in 20% (vol.) ethanol was sprayed from a positive capillary, and 96% ethanol was sprayed from a negative one. According to previous studies [32], the resulting aerosol is partially charged, so the chamber was covered with a grounded metal foil and closed with a metal mesh on top (Figure 5).

During the experiment, the mice were located in a foil-covered plastic container, closed on the top with a grounded conductive mesh, and placed in a sealed plastic bag with nozzles for aerosol inlet and outlet. The aerosol concentration measurements were made by taking a sample of the nanoaerosol from the inlet of the mice chamber.

### 4.3. Estimation of BLM NAP Concentration in Aerosol

Estimation of BLM NAP concentration was performed by the size distribution of aerosol particles. Spectra were obtained using a scanning mobility particle sizer, HCT SMPS (HCT Co., Ltd., Icheon-si, Gyeonggi-do, Republic of Korea). The BLM NAPs spectrum was recorded 40 min after the start of the experiment. As can be seen from Figure 6, the particle size distribution in the range from 7 to 550 nm has a broad peak with a maximum of ~100 nm. The size distribution of glucose NAPs is presented in Appendix A.

Since the concentration of the generated aerosol can vary with time, the average concentration of BLM NAP aerosol for a certain period of time was determined using nylon nanofilters fabricated by electrospinning and retaining >99% of ~100 nm particles [52]. The aerosol was collected on a nanofilter during the entire experiment at a rate of 0.1 L/min through a special outlet before the aerosol entered the chamber (Figure 6). After the experiment, the filter holder was opened, and the filter was separated from the support meshes and placed in a 0.5 mL tube. Then, BLM from each filter was extracted once with 100 μL of water with vigorous stirring on a vortex. The mass of BLM in the washout was determined using a NanoPhotometer P330 (Implen, Munich, Germany) at a wavelength of 294 nm using the absorption coefficient of BLM sulfate E^1%^ = 121.5 [Merck Index, 11th ed., 201–202, #1324].

The fraction of aerosol deposited in the lungs was determined by the Multiple-Path Particle Dosimetry Model (MPPD) [34,35], using the same breathing parameters as in [30], namely, the breath frequency n = 160 min ^−1^ and the tidal volume v = 0.2 mL. According to the MPPD model, the mass fraction of aerosol deposited in the lungs is ω = 35%. The average dose received by a mouse per day was calculated by the formula
D = n × v × t × ω × d × C_m_(1)
where t is the total exposure time for one day, and d is the correction for the deposition of aerosol in the chamber, equal to 0.8 (see Results Section for details). 

### 4.4. Animals and Models of Lung Metastasis

C57BL/6 6–8-week-old male mice were obtained from the animal breeding facility of the N.N. Blokhin Russian Cancer Research Center. All animal experiments were performed in accordance with Russian law and were approved by the Ethics Committee of the N.N. Blokhin Cancer Research Center.

Mice were injected subcutaneously (s.c.) in the right flank with a 10% tumor tissue suspension of murine Lewis lung carcinoma (LLC) or melanoma B16F10, resuspended in serum-free 199 medium in a final volume of 0.4 mL. When the tumor size reached 5 × 5 mm (4–5 days after tumor implantation), mice were randomized into treatment/control groups and treated with i.p. injections of BLM or exposed to BLM nanoaerosol. Control groups of animals received 0.9% NaCl in equivalent volumes. Body weight and tumor volumes were measured twice weekly, and calculated by the following formula: π/6 × length × width × height (mm^3^). Antitumor effect was estimated by tumor growth inhibition (%TGI) and increase in the lifespan (%ILS) of animals. 

Exposure of mice: Ten mice were placed inside a plastic chamber as described earlier [38] and exposed to 0.5% BLM NAPs daily for 2.5–5 h over 14 days. The total dose of BLM was 0.22–0.6 mg/kg per mouse. Mice of the control group were exposed to 1% glucose NAPs in equivalent conditions.

#### 4.4.1. Pulmonary Fibrosis Model

Fibrotic injuries were induced in mice by BLM. Briefly, mice were anesthetized with i.m. injections of 80 mg/kg Zoletil 100 (Carros, France) and 10 mg/kg Xylazine (Berlin, Germany) and given an intratracheal instillation of 5 mg/kg BLM. Lungs were harvested on day 15 after treatment. The slices were analyzed with a microscope using a ×10 objective. The slices were chosen at five random fields (×100 magnification) by slide for each of ten animals per group.

#### 4.4.2. Histology

Mice were euthanized by an overdose of diethyl ether by inhalation and autopsied for examination and weighing of spleens, lungs, and livers. Lungs were fixed in 10% buffered formalin at room temperature for 24 h. The tissues were dehydrated gradually in ethanol, embedded in paraffin, cut into 5–6 μm sections, stained with hematoxylin and eosin (H&E), and analyzed under light microscopy (Carl Zeiss, Jena, Germany) by a pathologist blinded to the treatments.

The number of macroscopically visible pulmonary metastatic nodules per mouse was counted. Lung metastases were counted manually under a light microscope using a micrometer in five H&E-stained sections.

### 4.5. Liquid Chromatography and Tandem Mass Spectrometry (LC-MS/MS)

#### 4.5.1. Sample Preparation

The calibration curve for BLM forms A2 and B2 were built in digested mouse plasma as a surrogate matrix. The concentration range of BLM used to obtain the calibration curve was 0–2000 ng/mL. The scheme for preparation of the BLM solutions for the quality-control calibration samples was the same as that for blood plasma samples as described below and was adapted from Kosjek et al. [53]

The plasma samples were quickly thawed, and 40 (20) µL aliquots were diluted to 600 µL with 0.1% formic acid and filtered through 0.45 mm cellulose acetate syringe filters. SPE of the plasma samples was performed using a CommaSep HLB plate (Copure, Guixi, China). The wells were preconditioned with 600 µL of 100% LC-MS-grade methanol and equilibrated with 600 µL of 0.1% formic acid. After the sample filtrates were loaded to the wells, the sorbent was washed 3 times with 600 µL of MilliQ water and dried under vacuum for 30 min and subsequently eluted using 100 µL of MilliQ water/MeOH (6/4) and with 50 µL acetonitrile. Eluates were used for MS analysis.

#### 4.5.2. LC/MRM-MS

All samples were analyzed in duplicate with an HPLC-MS system consisting of an ExionLC™ (UHPLC system (ThermoFisher Scientific, Waltham, MA, USA) coupled online to a SCIEX QTRAP 6500+ triple quadrupole mass spectrometer (SCIEX, Toronto, ON, Canada). LC-MS parameters, such as the LC gradient and the MRM parameters (Q1 and MRM scans) were adapted and optimized based on previous studies [53]. 

The sample volume loaded was 10 μL per injection. HPLC separation was carried out using an InfinityLab Poroshell 120 HILIC-Z (HILIC, 2.1 mm × 150 mm, 1.9 µm) (Agilent, Santa Clara, CA, USA) with gradient elution. Mobile phase A was 10% of 200 mM ammonium formate (pH = 3) formic acid in water; mobile phase B was 10% of 200 mM ammonium formate (pH = 3) in acetonitrile. LC separations were performed at a flow of 0.6 mL/min using a 10 min gradient from 100 to 60% of mobile phase B. Mass-spectrometric measurements were carried out using the multiple reaction monitoring (MRM) acquisition method. The electrospray ionization (ESI) source settings were as follows: ion spray voltage 4000 V, temperature 450 °C, ion source gas flow 40 L/min.

Two BLM forms, A2 and B2, were detected and analyzed in MRM experiments with the corresponding transition list (Q1/Q3) masses: BLM-A2 (*m*/*z* 708/551 and 708/399) and BLM-B2 (*m*/*z* 713/624 and 713/598) at the retention time (RT) of 3.5 min. Optimization of collisional energy (CE) was performed for each transition with 10 V step from 10 to 70 V. The optimal values were 50 and 60 V for BLM-A2 (*m*/*z* 708/551 and 708/399) and 50 V for both transitions for BLM-B2 (*m*/*z* 713/624 and 713/598).

The most intensive signal which was used for plasma sample analysis was observed for BLM-A2 with transition (Q1/Q3)—*m*/*z* 708/399 and was further used for plasma sample analysis.

### 4.6. Data Analysis

Skyline Quantitative Analysis software (version 20.2.0.343, University of Washington) was used for LC-MRM/MS quantitative analysis. The calibration curve was generated using 1/(x × x)-weighted linear regression methods to calculate the peptide concentrations in the measured samples (Appendix A).

Statistical processing was carried out using the SPSS 21 package. The Mann–Whitney–Wilcoxon test was used to compare quantitative traits in groups. For survival analysis, Kaplan–Meier curves were constructed and compared using a log-rank test. The results were considered statistically significant at *p* < 0.05.

## 5. Conclusions

The dose-dependent effect of aerosolized BLM was studied in comparison with the intraperitoneally treatment using two murine cancer models. The 100 nm particles were formed using a nanoaerosol generator with gas-phase neutralization. It was found that low long-term inhalation of BLM in nanoaerosol form demonstrates bioavailability similar to intraperitoneal treatment. According to the pharmacokinetics study, BLM plasma levels were 100 times lower compared with thos of intraperitoneal injection. However, long-term exposure to nanoaerosol is still capable of suppressing cancer cell growth. The lung histology analysis for the mice treated with BLM NAPs at doses of about 0.35 mg/kg demonstrated only focal-diffuse infiltration of the lung parenchyma by interstitial macrophages and mononuclear cells. Thus, our results show that nanoparticle delivery by nanoaerosol inhalation is promising for tumor suppression.

## Figures and Tables

**Figure 1 molecules-28-04157-f001:**
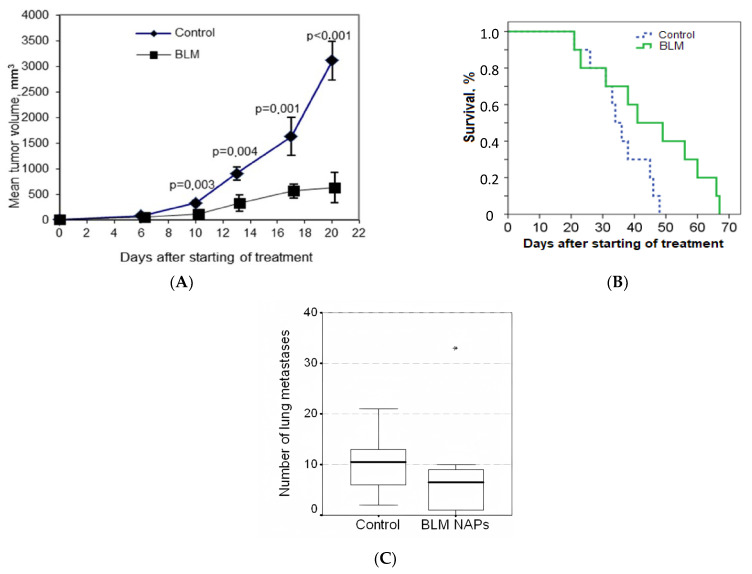
Effect of bleomycins nanoparticles (BLM NAPs) on murine melanoma growth and metastasis in vivo. C57BL6 mice were inoculated s.c. on the right flank with 0.5 mL of 10% B16F10 melanoma suspension. The mice were exposed to BLM NAPs or glucose NAPs (control) for 5 h daily for 14 days. (**A**) Mice inhaled with BLM NAPs had significantly smaller tumor volumes compared with control mice beginning from 10 to 20 days after the start of treatment. (**B**) The survival of mice in the BLM NAP group is 26% higher than in the control group. (**C**) The number of lung metastases in mice in the control and experimental groups. Error bars represent the standard errors of the mean (SEM).

**Figure 2 molecules-28-04157-f002:**
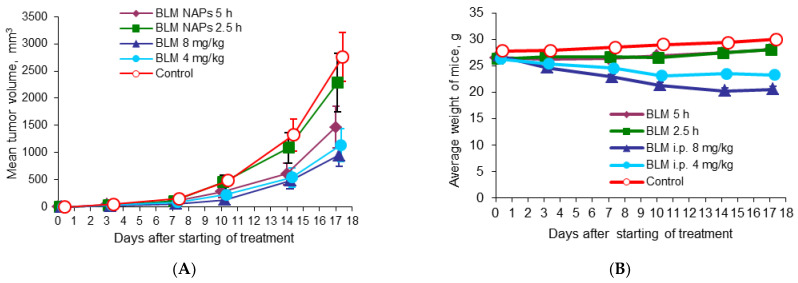
Effect of BLM NAPs on murine melanoma B16F10 (am) growth in vivo. C57BL6 mice were inoculated s.c. with 0.5 mL of 10% B16F10 melanoma tissue suspension. The treated groups were exposed to BLM NAPs for 5 or 2.5 h or BLM was injected i.p. 8 or 4 mg/kg daily for 14 days. (**A**) Mice treated with intraperitoneal BLM had significantly smaller tumor volumes compared with control mice. BLM NAPs induced unsubstantial TGI. (**B**) Mice treated with i.p. BLM had lower body weights compared to mice of control or BLM NAPs groups. Error bars represent the standard errors of the mean (SEM).

**Figure 3 molecules-28-04157-f003:**
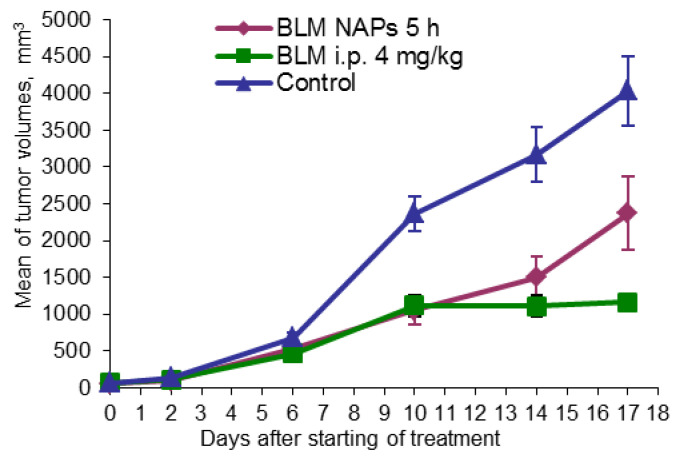
Effect of BLM NAPs on murine lung carcinoma LLC growth in vivo. C57BL6 mice were inoculated s.c. on the right flank with 0.5 mL of 10% LLC tissue suspension. The treated groups of mice were exposed with BLM NAPs for 5 days or were injected with 4 mg/kg BLM i.p. daily for 14 days.

**Figure 4 molecules-28-04157-f004:**
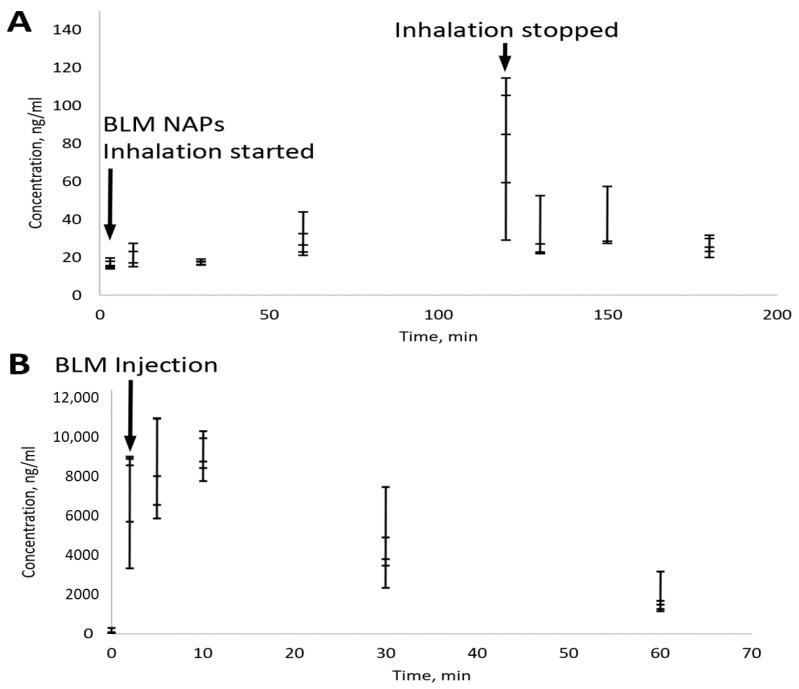
Pharmacokinetics of BLM delivered by nanoaerosol (**A**) and i.p. injection (**B**) measured in serum. Bars indicate standard error for a group of five mice. (**A**) Total BLM dose was 300 ng per mouse. (**B**) The administered BLM dose was 4 mg/kg.

**Figure 5 molecules-28-04157-f005:**
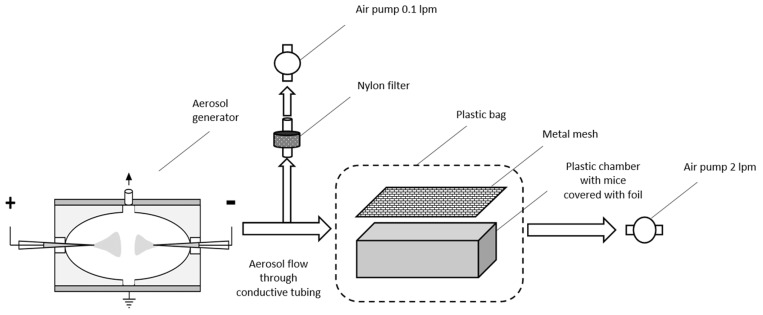
A schematic illustration of nanoaerosol generator and exposure chamber used in the experiments on inhalation of BLM nanoaerosol.

**Figure 6 molecules-28-04157-f006:**
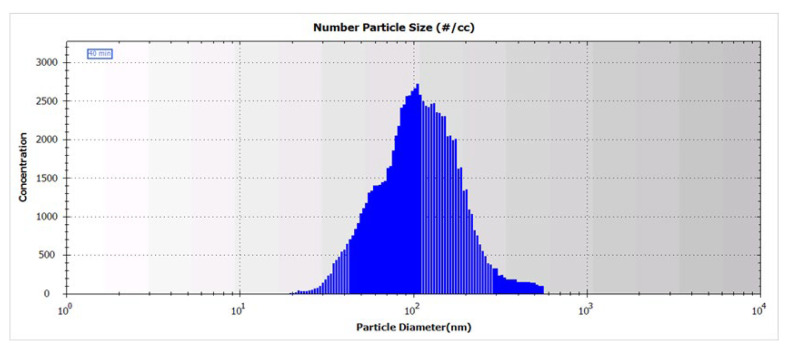
Size distribution of BLM NAPs in aerosol generated by spraying of 0.5% solution of BLM sulfate in 50% ethanol.

**Table 1 molecules-28-04157-t001:** Calculation of the inhaled dose of BLM for all experiments. Error values in aerosol concentration show standard deviation of the mean values.

Model	Inhalation Time, h/Day	Number of Days	Average Aerosol Concentration, μg/L	Total Dose, μg/Mouse	Total Dose, mg/kg
B16F10 melanoma (experiment 1)	5	14	0.23 ± 0.11	8.5 ± 4.2	0.35 ± 0.17
B16F10 melanoma (experiment 2)	5	14	0.19 ± 0.13	7.1 ± 4.7	0.27 ± 0.18
B16F10 melanoma (experiment 3)	2.5	14	0.31 ± 0.17	5.8 ± 3.2	0.22 ± 0.12
Lewis lung carcinoma	5	14	0.43 ± 0.20	15.9 ± 7.4	0.60 ± 0.28

**Table 2 molecules-28-04157-t002:** Hematological toxicity of BLM administration: absolute weights of spleen after i.p. and aerosol administration and in control group.

Groups	Absolute Weight of Spleen, g
Control	0.226 ± 0.048
BLM NAPs 5 h	0.214 ± 0.050
BLM NAPs 2.5 h	0.182 ± 0.007
BLM i.p. 8 mg/kg	0.107 ± 0.011 *
BLM i.p. 4 mg/kg	0.104 ± 0.008 *

Notes: Data are mean ± SEM of ten mice per group. * Significantly different from control (*p* < 0.05).

## Data Availability

All the data are available in Appendix A.

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
