# Peer review of "Antitumor Effect of Bleomycin Nanoaerosol in Murine Carcinoma Model"

_molecules, 2023, doi:10.3390/molecules28104157_

Round 1

Reviewer 1 Report

This paper reports on a comparative study on the antitumor effect of bleomycin nanoaerosol administration and bleomycin i.p. injections in murine carcinoma models. The result of this study is of interest.

Comments:

1.       In-text citations should be double-checked. For example, ref. 31 (line 69) is irrelevant to the corresponding discussion. Also, ref. “S1” (line 189) should be corrected.

2.       I am wondering why glucose has been used for treating control animals. It is well understood that glucose induces tumor growth.

3.       Healthy animals should be also exposed to the BLM nanoaerosol for evaluating the biochemical effects.

4.       The aerosol formation condition should be optimized and the results should be discussed in the main text.

Author Response

This paper reports on a comparative study on the antitumor effect of bleomycin nanoaerosol administration and bleomycin i.p. injections in murine carcinoma models. The result of this study is of interest.

Answer: We appreciate the Reviewer’s comments and positive feedback!  We have done our best to improve the text and address all the comments and criticism.

Comments:

  1. In-text citations should be double-checked. For example, ref. 31 (line 69) is irrelevant to the corresponding discussion. Also, ref. “S1” (line 189) should be corrected.

Answer: We apologize for reference numbers errors. Corrected throughout the text.

  1. I am wondering why glucose has been used for treating control animals. It is well understood that glucose induces tumor growth.

Answer: The aerosol concentration of glucose in the control experiments is 0.34 µg/l. After 5 hours of exposure single mouse will receive an aerosol dose of 2.6 µg. Even with a one-time injection of this dose into the blood, the change in glucose concentration will be 0.009 mM. But according to studies of the effect of hyperglycemia on cancer progression, changes in tumor growth are observed after much larger changes in glucose concentration (by 20 mM and more). [Li, W., Zhang, X., Sang, H. et al. Effects of hyperglycemia on the progression of tumor diseases. J Exp Clin Cancer Res 38, 327 (2019). https://doi.org/10.1186/s13046-019-1309-6].

  1. Healthy animals should be also exposed to the BLM nanoaerosol for evaluating the biochemical effects.

Answer: Previously, we specifically studied the effect of BLM nanoparticles on the lungs of healthy mice [Shlyapnikova EA et al. Exposure to bleomycin nanoaerosol does not induce fibrosis in mice. Eur J Nanomed. 2016;8(4):213–223, doi: 10.1515/ejnm-2016-0013.]. Inhalation of ~100 nm BLM aerosol particles for 13 days with a total dose of ~1 mg/kg did not cause pneumofibrosis in lungs in contrast to bolus instillation of similar doses; only pneumonitis and infiltration of macrophages in all parts of lungs were observed. These effects were notably reversible and disappeared after two weeks without aerosol inhalation. We have corrected the reference to this study in the text of Introduction and Discussion.

  1. The aerosol formation condition should be optimized and the results should be discussed in the main text.

Answer: The basic design, efficiency and optimization of the nanoaerosol generator performance used in the study were described in detail in our previous paper [Morozov VN et al. Generation and delivery of nanoaerosols from biological and biologically active substances. J. Aerosol Sci. 2014; 69, 48–61]. It has been also demonstrated that biological substances retain their structure and functional activity upon such atomization process and that negligible amounts of reactive oxygen species accompany the process [Kanev et al. Are reactive species generated in electrospray at low current? Anal. Chem. 2014; 86:1511–7.]. We have added the references to these studies to the text of Introduction and Materials and Methods.

Reviewer 2 Report

1- in the introduction give more details about Bleomycin and the used technique

2- give a brief description about the e generator design

3-most of the references are old, so  increase number of references from the last 5 years 

Check the spellings, punctuation and grammatical errors throughout the manuscript.

Author Response

Answer: We appreciate the Reviewer’s comments and positive feedback!  We have done our best to improve the text and address all the comments and criticism.

1- in the introduction give more details about Bleomycin and the used technique

Answer: We have extended the Introduction text and the aerosolization technique description in Materials and Methods.

2- give a brief description about the e generator design

Answer: We have extended the nanoaerosol generator description in Materials and Methods and added a schematic illustration in Supplementary information.

3-most of the references are old, so  increase number of references from the last 5 years 

Answer: Nine references to aerosols applications and bleomycin-induced fibrosis studies from the last 5 years were added to the Introduction text.

Check the spellings, punctuation and grammatical errors throughout the manuscript.

Answer: We apologize for punctuation and grammatical errors. The text was revised.

Reviewer 3 Report

There are some errors that need to be fixed in the work I reviewed:

- a claim about the dispersion of quantum dots is made in section 1, lines 61–64: however, the source of the claim is not cited;

- in line 128 the article discusses the design of the experiment chamber and how the concentration of nanoparticles was determined using a nanofilter, with reference to figure 2: this is misleading, since it's figure 1 that depicts the structure. Figure 2 is actually a graph showing the concentration of different-sized particles, as can be seen in a further image reported in the article;

- in line 138 the author proposes again to refer to figure 2 to see how the amount of bleomycin deposited in the lungs was determined. This is inaccurate since, as previously stated, figure 2 concerns a different topic. Furthermore, the article's assertion is not described by any figure at all;

- the article incorrectly cites picture S1 in line 189 when discussing the calibration curve, because figure S1 depicts the size and concentration of glucose NAPs. 

Author Response

There are some errors that need to be fixed in the work I reviewed:

Answer: We appreciate the Reviewer’s comments and positive feedback! We apologize for typos and errors. The text was revised.

- a claim about the dispersion of quantum dots is made in section 1, lines 61–64: however, the source of the claim is not cited;

Answer: The corresponding reference was introduced into the Manuscript

- in line 128 the article discusses the design of the experiment chamber and how the concentration of nanoparticles was determined using a nanofilter, with reference to figure 2: this is misleading, since it's figure 1 that depicts the structure. Figure 2 is actually a graph showing the concentration of different-sized particles, as can be seen in a further image reported in the article;

Answer: Thank you! The reference to the figure was corrected.

- in line 138 the author proposes again to refer to figure 2 to see how the amount of bleomycin deposited in the lungs was determined. This is inaccurate since, as previously stated, figure 2 concerns a different topic. Furthermore, the article's assertion is not described by any figure at all;

Answer: We apologize for the typos. The confusing reference was removed.

- the article incorrectly cites picture S1 in line 189 when discussing the calibration curve, because figure S1 depicts the size and concentration of glucose NAPs. 

Answer: Thank you! The misleading reference was corrected.